# Kinetics and Mechanisms of *Saccharomyces boulardii* Release from Optimized Whey Protein–Agavin–Alginate Beads under Simulated Gastrointestinal Conditions

**DOI:** 10.3390/bioengineering9090460

**Published:** 2022-09-09

**Authors:** María Sady Chávez-Falcón, Carolina Buitrago-Arias, Sandra Victoria Avila-Reyes, Javier Solorza-Feria, Martha Lucía Arenas-Ocampo, Brenda Hildeliza Camacho-Díaz, Antonio Ruperto Jiménez-Aparicio

**Affiliations:** 1Department of Biotechnology, Centro de Desarrollo de Productos Bióticos, Instituto Politécnico Nacional, Carretera Yautepec-Jojutla, Km. 6, Calle CEPROBI No. 8, Colonia San Isidro, Yautepec C.P. 62731, Mexico; 2CONACyT-Instituto Politécnico Nacional, Centro de Desarrollo de Productos Bióticos, Carretera Yautepec-Jojutla, Km. 6, Calle CEPROBI No. 8, Colonia San Isidro, Yautepec C.P. 62731, Mexico

**Keywords:** probiotic, Agave, ionic gelation, cell release, Damköhler number

## Abstract

Encapsulation is a process in which a base material is encapsulated in a wall material that can protect it against external factors and/or improve its bioavailability. Among the different encapsulation techniques, ionic gelation stands out as being useful for thermolabile compounds. The aim of this work was to encapsulate *Saccharomyces boulardii* by ionic gelation using agavins (A) and whey protein (WP) as wall materials and to evaluate the morphostructural changes that occur during in vitro gastrointestinal digestion. Encapsulations at different levels of A and WP were analyzed using microscopic, spectroscopic and thermal techniques. Encapsulation efficiency and cell viability were evaluated. *S. boulardii* encapsulated at 5% A: 3.75% WP (AWB6) showed 88.5% cell survival after the simulated gastrointestinal digestion; the bead showed a significantly different microstructure from the controls. The mixture of A and WP increased in the survival of *S. boulardii* respect to those encapsulated with alginate, A or WP alone. The binary material mixture simultaneously allowed a controlled release of *S. boulardii* by mostly diffusive Fickian mechanisms and swelling. The cell-release time was found to control the increment of the Damköhler number when A and WP were substrates for *S. boulardii*, in this way allowing greater protection against gastrointestinal conditions.

## 1. Introduction

Encapsulation is a process by which bioactive compounds and/or cells are coated with another protective material or mixtures of protective materials, protecting the encapsulated material from specific conditions, such as oxygen, high acidity and gastric conditions [1,2,3,4,5]. There are different encapsulation techniques, such as extrusion, ionic gelation, emulsion, spray drying, spray cooling, fluidized bed, freeze drying, spray freeze drying, coacervation, electro-spraying, ultrasonic vacuum spray drying, immersion spray technology and the hybridization method [6]. Among these, ionic gelation stands out for being useful for thermolabile compounds, low-cost and easy to scale-up, generally using alginate as a gelling agent—a hydrophilic polysaccharide that is biocompatible, biodegradable, non-toxic and that gels in the presence of bivalent metal ions such as Ca^2+^ [6,7,8,9,10].

In the encapsulation of probiotic cells, the wall material is an important aspect to consider, as it must protect the cells during their passage through the gastrointestinal tract and have a controlled release. Among the compounds that can form this wall material can be found polysaccharides, lipids, proteins and their mixtures [11,12]. Different compounds give different conformations to the capsules, such as core type (mononuclear), multinuclear type (polynuclear) and matrix type [13]. 

The morphostructure and the release mechanism of the capsules can be modified to suit specific applications and release sites of bioactive compounds and/or cells [14]. Morphostructural analysis of the capsules provides information on the wall materials and cells, as well as their arrangement, allowing us to determine the relationship between the structure and function of the system [15,16,17,18]. The analysis of the release mechanism provides useful information that allows beads to be designed depending on the application, degree of bioavailability of the released biomaterial and specific release site.

In the study of the release of bioactive compounds/cells from the encapsulating matrix, three stages must be considered: (a) release from the surface, (b) diffusion through the swollen matrix and (c) erosion of the matrix [19]. Kinetic data on release behavior can be analyzed using theoretical, empirical or semi-empirical equations [20]. Some of the most commonly used equations for geometries that commonly correspond to capsules are [19]:

The Zero-Order model (Equation (1)) considers the released fraction to be independent of the initial concentration [19,20]:M*_t_*/M_∞_ = 100(1 − kt)… for M*_t_*/M_∞_ = kt(1)

Remarks: M*_t_*, amount released after time t; M_∞_, amount released at infinity or equilibrium, k, rate constant. Fraction released independent of initial concentration. 

Higuchi’s model (Equation (2)) is applicable to diffusion-controlled release through water-filled pores and can be affected by pH and temperature [19,21,22]:M*_t_*/M_∞_ = kt^0.5^…  for 0.1 < M*_t_*/M_∞_ < 0.6(2)

Korsmeyer–Peppas model (Equation (3)) applies when the system has more than one release mechanism [23]:M*_t_*/M_∞_ = kt^n^(3)

Remarks: For spherical particles, n ≤ 0.43 for Fickian diffusion (Case I transport); 0.43 ≤ n ≤ 0.85 for non-Fickian (diffusion or swelling); n ≥ 0.85 for a Case II transport; and n > 1 for a Super Case II transport.

Finally, the Peppas–Sahlin equation (Equation (4)) quantifies the relative contributions to the Fickian and relaxation transport [19,24]:M*_t_*/M_∞_ = k_d_t^m^ + k_r_t^2m^…(4)

Remarks: k_d_ t^m^ for diffusion and k_r_ t^2m^ for Case II transport.

Most mathematical modeling studies have been used only in pharmaceutical studies [19]. Regarding the bead, a morphostructural analysis provides more detailed information regarding the shape, size and arrangement of the various parts that make up the bead. In turn, it is sought that through the relationships between dimensionless parameters, such as the Damköhler number, chemical interactions, or indicators of biomechanical importance in the encapsulates can be identified, in order to provide an analytical solution that involves a reduced computational cost compared to a large-scale numerical calculation [25].

Wall materials such as whey protein (WP) and reserve polysaccharides such as agavins (A) have been shown to be favorable materials for the encapsulation of probiotic microorganisms and bioactive compounds [26,27,28,29]. Agavins have a prebiotic effect [30,31] and, in turn, have the potential to be incorporated either in a mixture or individually as wall material. These agavins are defined as a heterogeneous mixture of fructose polymers linked by fructose–fructose glycosidic bonds [32,33]. Due to the β-(2→1) and β-(2→6) structure and bond type of these molecules, they have been recognized as prebiotic ingredients because they stimulate bacterial growth in the colon, which benefits gastrointestinal health and metabolic effects [34,35]. In addition, the branched structure of agavins may confer different functional properties than linear fructans such as inulin, such as allowing a higher water absorption capacity, which favors a greater internal plasticization of fructans and microstructural differences [36,37,38]. Whey proteins are a mixture of proteins with numerous and diverse functional properties and, therefore, may have many potential uses. The main whey proteins are β-lactoglobulin and *α*-lactoalbumin, which represent approximately 70% of total proteins and are responsible for the hydration, gelling and surfactant properties of protein ingredients [39]. Several authors recommend the use of prebiotics as wall material when encapsulating probiotic microorganisms, as they improve their protection [29]. Agavins and whey proteins have been used as spray-dry wall materials, both individually or in combination, to encapsulate probiotics such as *Enterococcus faecium*, *Lactobacillus acidophilus*, *Bifidobacterium pseudocatenulatum* CECT 7765, *Bifidobacterium bifidium* and *Saccharomyces boulardii*. The mentioned materials improve the encapsulation rate, viability during storage and simulated gastrointestinal testing, producing more stable beads with potential application in different foods, as they significantly preserve the quality of food matrices [26,27,40,41,42]. The FAO/WHO (2001) [43] defines probiotics as “live microorganisms that, when consumed in adequate amounts, confer a healthful effect on the host”. It is necessary that, when administered, probiotics have a concentration of 10^6^–10^7^ CFU g^−1^ at the time of consumption [44] and that they are able to colonize and maintain metabolic activity in the human intestinal tract [45]. *Saccharomyces boulardii* is a yeast considered a probiotic, offering protection against antibiotic-induced diarrhea, ulcerative colitis and Crohn’s disease [46]. It is relatively tolerant to acid pH and bile salts up to 0.3% (*w*/*w*), as well as to the human body temperature of 37 °C. This microorganism is usually dispensed in lyophilized form in soft gelatin beads, while its use in food matrix processing is scarce [47]. The prebiotic properties of agavins were proved using the probiotic microorganism *S. boulardii* as a model, which used agavins as a carbon source, where a greater growth of the population density was observed when such compounds were added to the culture medium [42]. Previous studies have shown that the encapsulation of *S. boulardii* using different techniques (spray drying, extrusion/cold gelation and layer-by-layer technique) seemed to have potential as an oral delivery system in pharmaceutical or food applications. In turn, it benefits both the survival and the bioavailability of *S. boulardii* within beads, making it more effective in the prevention and treatment of gastrointestinal diseases [26,27,48]. Although there is information on the advantages of encapsulating probiotics using fructans and whey protein as wall materials, it is not yet known how these components interact to conform the bead structure. A “desirable” structure not only protects the encapsulated material but also ensures the release of materials from the encapsulated core at specific targets and rates [49]. The study of the bead’s internal and external microstructure by digital image analysis and the possible release mechanism will allow the design of capsules according to their application, degree of bioavailability and specific release site. Therefore, the aim of the present work was to elucidate the effect of the wall materials on the internal and external conformation of bead microstructures and its relationship with cell-release mechanisms during in vitro gastrointestinal digestion of *S. boulardii* capsules obtained by ionic gelation using the response surface methodology.

## 2. Materials and Methods

### 2.1. Materials

The agavins were obtained in powder form following the Mexican patent MX/a/2015/016512 (Modular system and process for obtaining different products from agave fructans). Sodium Alginate REASOL^®^, molecular weight 216 g/mol, purity (95–100%). Whey protein was purchased from General Nutrition Center (GNC, Mexico) with the following components: protein (80%), ash (5%), lactose (4%) and fat (4%). Lyophilized yeast cells were obtained from Floratil *Saccharomyces boulardii* CNCM-I 745 beads (Biocodex, France). Yeast extract on peptone dextrose agar (YPD agar) was obtained from Sigma-Aldrich Química S. de R.L. de C.V., containing agar (10 g/L), bacteriological peptone (20 g/L), glucose (20 g/L) and yeast extract (10 g/L).

### 2.2. Preparation of Standard Inoculum for Encapsulation 

Lyophilized *Saccharomyces boulardii* cells were reactivated three times; they were first poured into 50 mL of sterile YPD broth and incubated for 24 h at 37 °C. Subsequently, a flask containing 50 mL of YPD broth was inoculated with the resulting medium at 5% and incubated for 12 h at 37 °C. Finally, a flask with 30 mL of YPD broth was inoculated with 2% of the resulting medium and incubated for 12 h at 37 °C [50,51]. The last reactivation was used to recover the microorganism to carry on the encapsulation. The optical density was monitored using a UV–visible spectrophotometer (Shimadzu, UV-1800, Japan) at a wavelength of 640 nm.

### 2.3. Solution Preparation of Agavins and Whey Protein

Dispersions of agavins and whey protein were made by mixing the powders in distilled water at neutral pH and room temperature (25 ± 2 °C) and stirring for 30 min with a magnetic stirrer until complete dissolution. Both dispersions were stored for 24 h at 4 °C to achieve complete hydration. Dropwise addition of 1 M NaOH or 1 M HCl solution was used to obtain samples at the required pH range [27,50].

Simultaneously, 30 mL solution of 0.96% alginate was prepared in a 50 mL Erlenmeyer flask. Then, the dispersions of agavins, whey protein and the total cell concentrate recovered from 30 mL of YPD broth were carefully added. Everything was mixed with a magnetic stirrer at 25 ± 2 °C until complete homogenization.

### 2.4. Encapsulation of Yeast by Ionic Gelation

To determine the concentrations of the wall materials of the capsules, a completely randomized experimental 3^2^-factorial design was used, with three replicates for each of the nine resulting treatments (Table 1), taking as response variable the percentage of encapsulation efficiency (%EE). Three test levels were considered: low (−1), intermediate (0) and high (1) levels for the concentrations of agavins (2.5, 3.75 and 5%) and whey protein (2.5, 3.75 and 5%) [29]. The seed culture was centrifuged (Hermle Z383 K centrifuge, Germany) at 10,000 rpm at 4 °C for 10 min, and the cell concentrate was washed twice with PBS solution (phosphate-buffered solution, pH = 7). Twelve types of beads were prepared with three replicates: beads of alginate (B), agavins (AB), whey protein (WB) as control beads, and the remaining nine beads were prepared based on a 3^2^-factorial design (Table 1). Encapsulation was performed by ionic drip gelling with a 5 mL syringe (21 G × 32 mm). The droplets were injected into a 0.2 M CaCl_2_ solution; they were allowed to stand for 30 min to harden and strengthen the crosslinking. Subsequently, the beads were recovered with a filter paper (Wathman No. 4) under sterile conditions [52,53].

### 2.5. Viability of Encapsulated Saccharomyces boulardii

The viability of yeast cells was quantified by %EE, which is defined as the detected concentration of the incorporated material in the formulation with respect to the initial concentration used [54]. It was calculated by means of Equation (5):%EE = (W_t_/W_i_) × 100…(5)
where %EE = percentage of encapsulation efficiency, W_t_ = detected concentration of the incorporated material (CFU), and W_i_ = initial concentration (CFU).

### 2.6. Optimization of the Morphostructure of Beads

#### Morphostructural Parameters by Microscopy

Micrographs were obtained from wet beads using a stereo optical microscope (Nikon Eclipse 50 i, Nikon, Japan), a fixed digital camera (Nikon digital Sight DS-2 mV, TV0.55 lens; Nikon, Japan) and MetaMorph Software (Version 6.1, 1992–2003). Thirty micrographs (with reflected light and 3X magnification) were obtained for each one of the five types of beads that showed a significant difference in terms of %EE and were stored in color *.tiff format. Those micrographs were subjected to digital image analysis using ImageJ software (v1.53s). The bead images were converted to grayscale (8 bits) and binarized (black and white) using the *Threshold* function. Those were manually adjusted to a grayscale range. Using these bead images, size (area and perimeter) and shape (circularity and solidity) parameters were analyzed. Morphostructural parameters of size and shape were evaluated in order to identify which beads were significantly influenced by the effect of different concentrations of wall materials (agavins and whey protein). A response surface analysis was then performed for the parameters described above, using the Minitab Graphs software (Minitab^®^ 18.1).

To evaluate the distribution of *S. boulardii* inside the wet beads, a laser scanning confocal microscope (Carl Zeiss Model CLSM 800) with Smartsem 5.6 software was used to obtain the micrographs. Two fluorochromes were used: propidium iodide and acridine orange. Five micrographs were taken on the central zone of each bead type (virtual zoom 0.5 = 40X) and were stored in color *.tiff format. They were subjected to digital image analysis using ImageJ software (v1.53s). Using the *Color Threshold* function, images were manually adjusted to a color scale to specifically select yeasts. The yeast images were binarized, and the following parameters were obtained using the *GLCM* (gray level co-occurrence matrix) texture plug-in: *second angular momentum*, *contrast*, *correlation* and *entropy*. Subsequently, the images were skeletonized using the *Skeletonize* tool. *Fractal dimension* and *lacunarity* parameters were obtained from these images using the *Frac Lac* plug-in. This skeletonization was performed based on the distribution of *S. boulardii* in the bead, since it has been observed that the cells tend to deposit in the crosslink structure of the wall materials [55,56]. This suggests that skeletonization provides indirect information about the *Beads Internal Networking (BIN)* or *mesostructure*, formed both individually and in combination by the alginate, agavins and whey protein.

The external morphology and texture of wet beads, called *Beads External Surface (BES)*, were observed using a Carl Zeiss EVO LS 10 scanning electron microscope (Life Science; Germany) with Zeiss Efficient Navigation Software (2.3 Blue Edition) at 40X and 500X optical magnification with backscattered electrons in the environmental mode. Five micrographs were taken for each type of bead (40X magnification). Meanwhile, micrograph sections were taken in different areas (top–bottom, left–right, center) of the beads and were stored in color *.tiff format. Micrographs were subjected to a digital image analysis using ImageJ software (v1.53s). The different micrograph sections were converted to grayscale (8 bits). Based on these, the following parameters were analyzed using the *GLCM* texture plug-in: *second angular momentum*, *inverse differential momentum*, *contrast*, *correlation and entropy*. Subsequently, the *SDBC* (Shifting Differential Box Counting) plug-in was used to obtain the value of the parameter *texture fractal dimension*.

### 2.7. Physicochemical Properties of the Capsules

Thermal profiles were obtained using differential scanning calorimetry equipment (TA Instruments, DSC Q20, USA) under a nitrogen atmosphere. Each sample was heated from 0 to 300 °C at a rate of 5 °C/min. All determinations were made in triplicate. The equipment was preliminarily calibrated with an indium standard reference [50]. Fourier transform infrared spectroscopy (FT-IR) was performed using a spectrophotometer (IRAffinity-1, Shimadzu, Japan) with an ATR (Attenuated Total Reflection) accessory with a zinc selenide crystal. For each spectrum, an average of 16 scans were recorded with a resolution of 8 cm^−1^ in the range of 400–4500 cm^−1^. The determinations were performed in triplicate.

### 2.8. Survival of Saccharomyces boulardii under In Vitro Gastrointestinal Digestion Conditions

#### 2.8.1. Viability

The different capsules were subjected to human gastric simulation according to the INFOGEST protocol [57] under the following conditions: 37 °C, 90 rpm. For the oral phase, the protocol included a 1:1 mixture of simulated salivary fluid (SSF), amylase, 2 min, pH 7. For the gastric phase, the protocol included a 1:1 mixture of simulated gastric fluid (SGF), pepsin, gastric lipase, 2 h, pH 3, and for the intestinal phase, a 1:1 mixture of simulated intestinal fluid (SIF), pancreatin, bile salts, 2 h, pH 7. A laser scanning confocal microscope (Model CLSM 800 Carl Zeiss) was used with the software Smartsem 5.6. to obtain micrographs of the beads at the beginning of the in vitro gastrointestinal digestion process (0 min) and at the end of the process (250 min). Two fluorochromes were used: propidium iodide and acridine orange. Viable cell counts were performed based on the plate count method on YPD agar and were indicated as CFU mL^−1^. First, 100 mg of beads were taken, and 10^−1^ to 10^−5^ dilutions were made in 1.5 mL Eppendorf tubes with 900 μL of 0.9% Sodium Citrate Sigma-Aldrich^®^, molecular weight 294.10 g/mol (mixing homogeneously with a vortex until complete dissolution of the beads). Subsequently, 10^−3^–10^−5^ dilutions were seeded in Petri dishes (mixing homogeneously with a vortex) using the microdroplet method proposed by Miles and Misra [58]. They were incubated for 24–48 h at 37 °C, and plate counts were performed. 

#### 2.8.2. Kinetics and Release Mechanisms

Five grams of beads were weighed and added in SSF, SGF and SIF release solutions [57]. The concentration of cells in the solutions was monitored in triplicate at 0, 1, 2, 5, 20, 35, 65, 95, 125, 130, 145, 160, 190, 220 and 250 min. Cells released (CFU mL^−1^) into the medium were quantified using a spectrophotometer at a wavelength of 640 nm, while viable cell counts were quantified according to the plate count method on YPD agar and measured as CFU mL^−1^. The release of yeast cells was analyzed according to the Higuchi (Equation (2)), Korsmeyer–Peppas (Equation (3)) and Peppas–Sahlin (Equation (4)) models. In addition, the parameters in each equation were determined. Finally, the Damköhler number (Da) was determined by Equation (6):Da = *S. boulardii* growth rate/*S. boulardii* cell release rate(6)

### 2.9. Statistical Analysis

One-way ANOVA (analysis of variance) and Tukey’s multiple-comparison test (α < 0.05) were performed. For data analysis and illustration, the Minitab Graphs package (Minitab^®^ 18.1), IBM SPSS Statistics (Version 25.0) and Microsoft Excel version 16.62 (22061100) software were used for statistical analysis. 

## 3. Results

### 3.1. Growth of Saccharomyces boulardii 

The growth kinetics of *S. boulardii* was carried out in relation to time (h) and optical density (O.D.). It was observed that the peak of the logarithmic phase occurred at 12 h with an O.D. of 1.34 [27]. This was done to estimate the time required for *S. boulardii* cells to reach a concentration of 10^6^ CFU mL^−1^ and, therefore, the time required to proceed to perform the encapsulation process.

### 3.2. Viability of Encapsulated Saccharomyces boulardii

The bead with the highest %EE was AWB9 with 96.77%, while the AWB7 beads had the lowest efficiency of 89.27%. The bead types that had a significant difference compared to controls were: AWB3 (96.043%), AWB5 (93.901%), AWB6 (95.698%), AWB8 (94.053%) and AWB9 (96.775%). These capsules were used for subsequent analyses (Appendix A). 

Therefore, agavins had a significant influence on the protection of *S. boulardii* compared to whey protein, due to the fact that beads with intermediate and high percentages (3.75 and 5%, respectively) of agavins had the highest %EE.

### 3.3. Morphostructural Characterization of the Beads

#### 3.3.1. Morphostructural Optimization of the Beads

Morphostructural parameters were analyzed by stereo microscopy. According to Pedreschi et al. [59], the structural characterization of an object is mainly based on the measurement of geometric properties. For this reason, the following parameters were evaluated by stereo microscopy: *area* (number of square units covering the projected bead surface, or number of pixels in the bead), *perimeter* (length of the outline of a bead, or number of pixels around the boundary of each bead), *circularity* (measure of how close the bead shape is to a perfect circle) and *solidity* (the ratio between the bead area and the convex hull of the bead).

According to the statistical analysis, the parameters of size (*area* and *perimeter*) were significantly affected (Appendix A). The behavior of these size and shape (*circularity* and *solidity*) parameters with different concentrations of wall materials was observed using a response surface analysis (Figure 1) [60]. Figure 1a shows that, even when the highest percentages of agavins were present, the *area* did not reach its maximum value; instead, the highest percentage of whey protein presented the highest value of *area*. Similarly, Figure 1b shows that whey protein alone reached the maximum values of *perimeter*. In other words, it was whey protein that determined the size of the capsules compared to agavins. Moreover, whey protein affected the morphology of the capsules, producing more spherical beads with a smoother surface that in combination with polysaccharides, modifying the properties of the wall material as well as the particle size [29].

The effect on circularity varied with the concentration of wall materials since the same concentration for AB and WB presented a similar effect. Intermediate values of *circularity* (0.77–0.80) were obtained from concentrations of approximately 3.4% whey protein and 4% agavins (Figure 1c). However, agavins and whey protein had a similar influence on *circularity* behavior. The agavins showed high *solidity* values (0.98–0.99) from concentrations of 3.8%, while whey protein presented high *solidity* values from concentrations of 4.8%. (Figure 1d). Both agavins and whey protein showed similar effects for the *solidity* parameter.

In order to contain the greatest number of cells (*area*) while delimiting a smaller amount of space (*perimeter*) with *circularity* and maximum *solidity* (the structure should be as solid as possible), it was desired that the bead shape be as similar as possible to a circle. Therefore, a mathematical optimization of these parameters was carried out for the AWB3, AWB5, AWB6, AWB8 and AWB9 beads. *Area*, *circularity* and *solidity* were weighted as *maximizing* them, while *minimizing* was set as a target for *perimeter*, and the degree of importance was set homogeneously. As a result of the mathematical optimization, three types of beads were obtained that met or were close to the previously established objectives: AWB5 (3.75% A: 3.75% WP), AWB8 (3.75% A: 5% WP) and AWB6 (5% A: 3.75% WP).

#### 3.3.2. Study of the Internal Morphostructure (Mesostructure) of the Beads

Micrographs of the three types of beads resulting from the optimization, together with B, AB and WB, were taken by means of laser scanning confocal microscopy. From the micrographs obtained, image analysis was performed using the ImageJ tool *Skeletonize* (Figure 2). Skeletonization extracts information about how the pixels are spatially related and defined to measure the length of each branch and the number of branches in each skeletonized feature from *BIN* [61,62]. 

Figure 2 shows that B had a mostly homogeneous and separated structure, whereas AB formed a mostly crosslinked network, while WB formed a network with less crosslinking. To test whether there was a significant difference between each structure as a function of agavins and whey protein concentration, texture parameters were evaluated. This evaluation was assessed with the *GLCM* texture plug-in as *second angular momentum* (a measure of the uniformity of the *BIN* texture), *contrast* (a measure of local variations in the *BIN*), *correlation* (a measure of linear dependence of intensity values in the *BIN*) and *entropy* (a measure of disorder in the *BIN*) [15,16,63]. Moreover, *fractal dimension* (compares how the detail of a pattern changes with the scale considered in the *BIN*) and *lacunarity* (a quantitative measure of the degree of clustering of the pore structure, representing the translational or rotational invariance in the *BIN*) were also assessed. Hristu et al. [16] and Smoczyński [63] mentioned that the parameters derived from the *GLCM* texture plug-in provide information about the spatial relationships between pixel intensities in the *Beads Internal Networking (BIN)*.

According to the parameters (Appendix A) *second angular momentum*, *contrast*, *correlation and entropy*, AWB5, which had equal concentrations of agavins and whey protein (3.75% A: 3.75% WP), showed a significant difference only compared to B and WB, but not AB. Therefore, it was deduced that agavins predominated these parameters, i.e., as they had a significant effect on the uniformity, variability, correlation and the disorder of the internal structure of the beads compared to B and WB. In the combination of agavins and whey protein wall materials, the complexity (*fractal dimension*) of the agavins predominated since the AWB5 was significantly different from B and WB. Thus, the agavins significantly influenced the complexity of the bead structure. The *lacunarity* presented a significant difference for AWB5 with respect to AB and, at the same time, similarity with B and WB. Therefore, for this parameter, it was the whey protein that had a significant influence. In summary, the agavins had a direct effect on the parameters of *second angular momentum*, *contrast*, *correlation*, *entropy and fractal dimension*. This means that agavins determined the internal structure of the beads compared to the whey protein.

#### 3.3.3. Study of the External Morphostructure of the Beads

Micrographs were taken by means of scanning electron microscopy, and image analysis was performed using ImageJ software (Figure 3). Parameters such as *second angular momentum*, *inverse differential momentum* (similarity of a pixel value in combination with all other neighboring pixel pairs in the *BES* images), *contrast*, *correlation*, *entropy and SDBC* (measures the fractal dimension of the *BES* texture) [64] were evaluated to obtain information regarding the texture of the *BES* (Appendix A). The *second angular momentum* and the *inverse differential momentum* of the AWB5 and AWB6 beads were significantly different from B, AB, WB and AWB8. The B, AB and WB beads did not present significant differences among them, either in these parameters or in the *contrast* parameter. Therefore, it was inferred that the wall materials of agavins and whey protein did not individually have a significant influence on the texture of the *Beads External Surface*; however, when combined, they did present a significant difference in their uniformity, since the AWB5 and AWB6 beads were more homogeneous.

Regarding *correlation*, the AWB5 and AWB6 beads significantly presented the highest values of correlation, compared to B, AB, WB and AWB8. For *entropy*, the beads that presented a significant difference with the lowest disorder value were AWB5 and AWB6 compared to all other bead types. The B, AB and WB beads did not present significant differences among them. The AWB5 bead presented a significant difference with the lowest tortuosity value compared to AB. Because it was found in equal concentrations in the wall material, this suggested that the whey protein governed the *SDBC* parameter. In general, the whey protein determined the external structure of the beads.

### 3.4. Physicochemical Properties of Capsules

#### 3.4.1. Conformational Analysis by Fourier Transform Infrared Spectroscopy (FT-IR)

Figure 4 shows the absorbance spectra obtained for the B, AB, WB and the AWB5, AWB6 and AWB8 beads. The FT-IR spectrum of B presented a peak at 1020 cm^−1^ assigned to C-O-C stretching groups and peaks at 1400 and 1600 cm^−1^ corresponding to asymmetric and symmetric stretching of COO- bonds. As for the band observed at 3250 cm^−1^, this was attributed to O-H stretching groups [65,66,67]. Regarding AB, a characteristic absorption band at 900–1100 cm^−1^ was identified. Santiago-García, et al. [68] described a peak at 900–1200 cm^−1^ for agavins-type fructans. Particularly, the 900–1200 cm^−1^ absorption bands in the carbohydrate region were attributed to the vibrations of the C-O-C group in the cyclic structures, indicating a significant carbohydrate content [69,70]. The WB showed two characteristic peaks belonging to amides, the first in a band of 1490–1570 cm^−1^ derived from the N-H bending vibrations of the amide bond and C-N stretching [27,71,72]. Krimm and Bandekar [71] and Kong and Yu [72] mentioned that the most sensitive spectral band characterizing the components that conform to the secondary structure of proteins was from 1600–1700 cm^−1^, so there was similarity with the WB bead, since it had a peak from 1570–1700 cm^−1^, a consequence of the C=O vibrations of peptide bonds and the direct relationship they possess with the secondary structural elements: *β*-sheets, *α*-helices, chain turns and random coils belonging to its main proteins, β-lactoglobulin and *α*-lactalbumin [27,71,72].

The FT-IR spectra (Figure 4) were able to identify a (weak) conformational change in the structure of AB and WB, due to the interaction between alginate–agavins and alginate–whey protein. A 2250–2350 cm^−1^ peak was present in AB, attributed to the vibration of the C≡N group (disubstituted alkyne, saturated, very weak, sometimes not visible), and the 2800–2990 cm^−1^ peak was a consequence of the symmetric stretching of the CH_3_ and CH_2_-O groups, where the latter peak was also found in WB [73]. 

The correlation coefficients (R) of the spectra were analyzed, where it was observed that AWB6 beads showed a higher correlation with AB (0.9620). In turn, AWB8 beads showed a higher correlation with WB (0.9572).

#### 3.4.2. Thermal Properties by Differential Scanning Calorimetry (DSC)

The thermograms of the beads produced by the alginate, agavins and whey protein, either individually or in combination, are shown in Figure 5. The fusion temperature of B was 164.180 °C, with an enthalpy of 253.650 J/g and an endothermic peak similar to that reported for alginate beads [74]. It refers to the temperature required for the complete fusion of the organic compounds that conform the sample and the energy that the melting transitions need in order to occur. AB presented an endothermic peak at 151.475 °C with an enthalpy of 203.400 J/g. Similar results by Espinosa-Andrews et al. [75] and Ignot-Gutiérrez et al. [76] were attributed to the fusion point. The WB bead showed an endothermic peak at 148.225 °C with an enthalpy of 105.460 J/g, a peak associated with the denaturation of its component proteins [29]. 

AWB6 presented an endothermic peak at 155.82 °C (Figure 5), which was attributed to heat-induced transitions occurring in the agavins and the whey protein [53]. Through the increased degree of crosslinking of alginate by the presence of agavins and whey proteins, the melting temperature (Tm) was shifted to a high temperature of control B beads, indicating enhanced thermal resistance. This suggested that this conformed microstructure restrained heat conduction [54]. 

The AWB5, AWB6 and AWB8 beads presented a shift of the endothermic peak due to the combination of alginate, agavins and whey protein in different concentrations, forming mixtures that provided properties different from those of the *pure* wall materials (with *S. boulardii*). The fusion temperature of the AWB5 bead was 138.860 °C (enthalpy: 203.810 J/g), while the AWB6 bead was 155.820 °C (enthalpy: 198.400 J/g), and finally, the AWB8 bead was 149.360 °C (enthalpy: 147.270 J/g).

### 3.5. Survival of Saccharomyces boulardii under In Vitro Gastrointestinal Digestion Conditions

#### 3.5.1. Viability of *Saccharomyces boulardii*

Micrographs of the different capsules (Appendix A) at the beginning of in vitro gastrointestinal digestion (minute 0) and the end of digestion (minute 250) showed that the AWB6 and AWB8 beads presented higher viability compared to B, AB and WB. The latter (B, AB and WB) showed a higher number of dead cells (stained red) in contrast to the live cells (stained green). According to Bank [77], propidium iodide and acridine orange fluorochromes are fluorescent markers for simultaneous visualization of live and dead cells. Figure 6 shows the release of cells into the medium over time (CFU mL^−1^), while the images represent the viable/dead cells at specific points (at the beginning and at the end) of in vitro gastrointestinal digestion. The results of this assay were verified by viability tests.

According to the results obtained with the plate count technique, the AWB6 beads (5% A: 3.75% WP) showed the highest %viability (88.539) compared to B (70.37%), AB (75.58%) and WB (72.66%), while the AWB5 (83.99%), AWB6 and AWB8 (86.26%) beads did not significantly differ from each other.

#### 3.5.2. Kinetics and Release Mechanisms of *Saccharomyces boulardii*

The kinetics of *S. boulardii* release was related to the concentration of yeast detected in the medium and the amount of live yeast inside the bead (Figure 7). Thus, B, AB and WB presented a point at which both processes intersected (equilibrium point). This point was associated with the time at which both inside the bead and in the medium had the same concentration of live yeast. This was associated with changes in the configuration of the polymeric chains that form the internal/external structure of the capsules. The B bead had an intersection at 130 min (beginning of the intestinal phase), the AB at 145 min (intestinal phase), and the WB at 170 min (intestinal phase). In contrast, AWB5, AWB6 and AWB8 did not present intersections in any of the phases, which indicated that the beads reached the colon in bead form, since they showed controlled-release behavior. The AWB5 beads were the closest to intersect at some point after the intestinal phase, followed by the AWB8 beads and ending with the AWB6 beads. Finally, this confirmed that the structure of the mixed beads retained the morphostructural properties that provided *S. boulardii* with protection and a controlled-release mechanism until it reached the target site.

Table 2 shows the indexes and coefficients obtained for three mathematical models. The selection of the release model for the different *S. boulardii* capsules was performed according to the coefficient of determination (R^2^) obtained, which allowed us to know to what extent the regression line best represented the model [78]. The Peppas–Sahlin model presented the highest R^2^ values (0.8538–0.9848) in all cases, so it was selected as the model that best described the release and transport of *S. boulardii* in the different types of beads. The B, AB, WB and AWB5 beads presented relaxation release mechanisms. In this case, the polymeric chains of the alginate changed their configuration during the intestinal phase and thus led to the release of the microorganisms. The AWB6 bead (5% A: 3.75% WP ) had the least % cell release (Appendix A) and therefore the highest %viability at the end of gastrointestinal conditions, with respect to the B bead, which had 100% cell release after the intestinal phase. Meanwhile, the AWB8 beads presented a Fickian diffusion transport mechanism. This result showed that yeast release was carried out through two mechanisms: diffusion and relaxation. The first represented a greater contribution to the release of *S. boulardii*, while the inclusion of the second allowed obtaining an adequate mathematical adjustment [21]. The release mechanism for this type of bead used the concentration gradient as the driving force [23].

For the Damköhler number (Da), the AWB5, AWB6 and AWB8 beads presented values Da > 1, thus showing that for these beads, the reaction rate (growth of *S. boulardii*) governed these systems. This means that they contributed significantly to the trapping of available *S. boulardii* in the bead. In contrast, the controls had values Da < 1, indicating that the rate of transport governed the system, i.e., more yeast was diffused from the system than was generated [25].

## 4. Discussion

The combination of agavins and whey protein with alginate using the ionic gelation technique had a significant influence in obtaining higher %EE (94–97%) compared to the controls (B, AB and WB) [43]. Agavins gave the beads heterogeneous internal structure (entropy and fractal dimension), improving the crosslinking of the polymeric network [79]. 

On the other hand, whey protein provided a smoother homogeneous external structure, with less contact surface and optimal size that ensured the highest number of microorganisms were encapsulated in the smallest space. Moreover, whey protein did not compromise the fracture of the internal/external microstructure of the bead, due to the cohesiveness that globular-type proteins impart to the network [27,80]. 

Through DSC and FT-IR analysis, it was found that the mixture of agavins and whey protein maintained and/or compensated the weak interactions that allowed increases in the melting temperature and the energy needed to generate a change in the properties of the compounds that formed the bead. 

The optimal type of mixture for encapsulation was AWB6 (5% A: 3.75% WP), which presented the highest %viability (88.54%, corresponding to 7.3 × 10^7^ CFU/mL) after in vitro gastrointestinal digestion compared to AWB5, AWB8 and the controls, thus ensuring that it reached the concentration necessary to be considered functional [43,81]. 

The bead type with the highest %viability (5% A: 3.75% WP) had a Fickian diffusion transport mechanism, probably as it used the concentration gradient as the driving force with a release mechanism controlled by the bead microstructure [23]. The mathematical model that best described the behavior of all bead types was the Peppas–Sahlin model.

The optimal agavins/whey protein mixture obtained capsules with a more crosslinked matrix. This may be due to its gelling strength developed by agavins and whey protein in an alginate matrix, which enhanced the hardness and flexibility of capsules [82] (entropy: 0.144, fractal dimension: 1.616, lacunarity: 0.034), generated a more homogeneous external structure (entropy: 7.534, SDBC: 2.348) and optimal size (area: 5.431 μm^2^, perimeter: 9.082 μm), and induced a Fickian release mechanism with both transport phenomena, diffusive and relaxation, simultaneously (mainly diffusive) according to the Peppas–Sahlin model [19,21], with value Da > 1 (1.121).

## 5. Conclusions

Currently, it is essential to understand the behavior of capsules through physical, chemical and morphostructural characterization, as well as mathematical modeling that allows correlating the size, shape, surface area, bioavailability of the biomaterial, encapsulation efficiency, etc., for the design and manufacture of large-scale capsules. The optimization of the mixture of wall materials in the capsules was possible to define through morphometric measurements at three different levels of observation of the beads (whole bead, bead external surface and bead internal networking). The use of mixtures of agavins and whey protein using ion gelation increased the survival of *S. boulardii* after gastrointestinal conditions due to the crosslinking promotion compared to the use of alginate alone or as a single-component encapsulation.

## Figures and Tables

**Figure 1 bioengineering-09-00460-f001:**
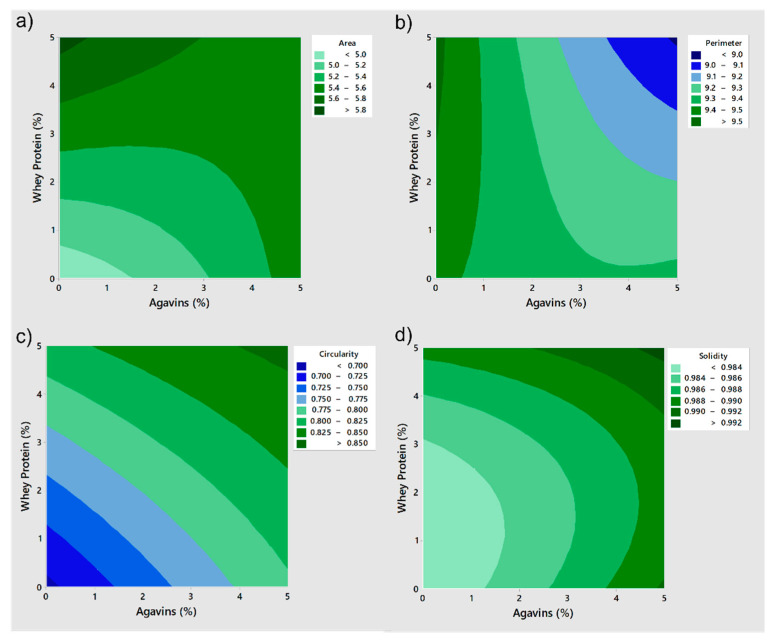
Contour plots: (**a**) Area vs. whey protein, agavins; (**b**) Perimeter vs. whey protein, agavins; (**c**) Circularity vs. whey protein, agavins; (**d**) Solidity vs. whey protein, agavins, obtained from a response surface analysis for each of the different types of beads with *S. boulardii*.

**Figure 2 bioengineering-09-00460-f002:**
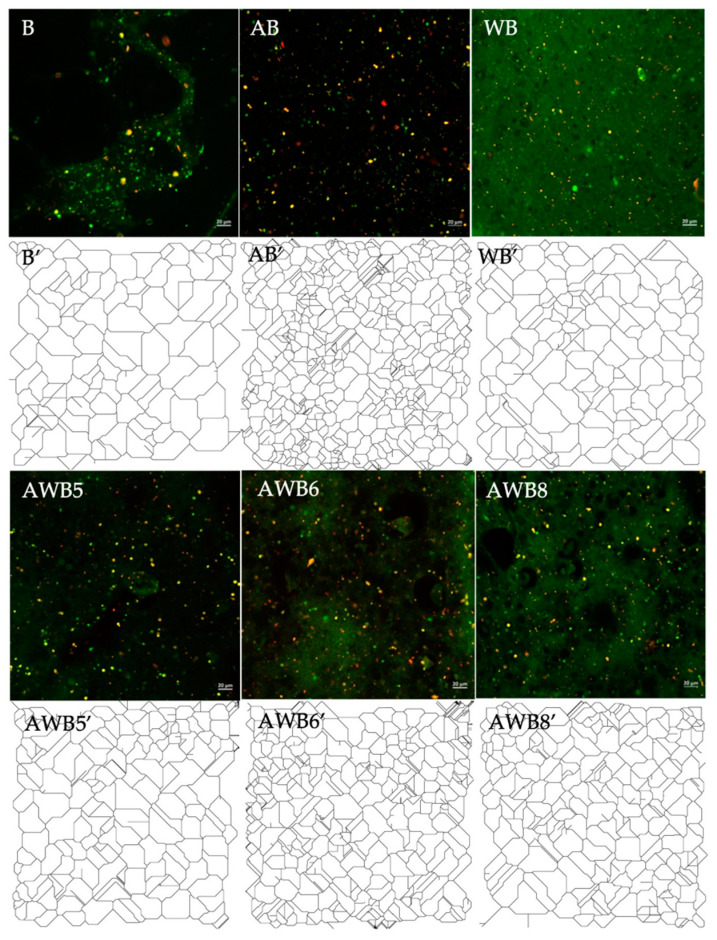
Micrographs obtained by laser scanning confocal microscopy (scale: 200 microns, zoom central part of the beads) of beads with *S. boulardii*. Skeletonization of micrographs obtained through Image J software (nomenclature with apostrophe).

**Figure 3 bioengineering-09-00460-f003:**
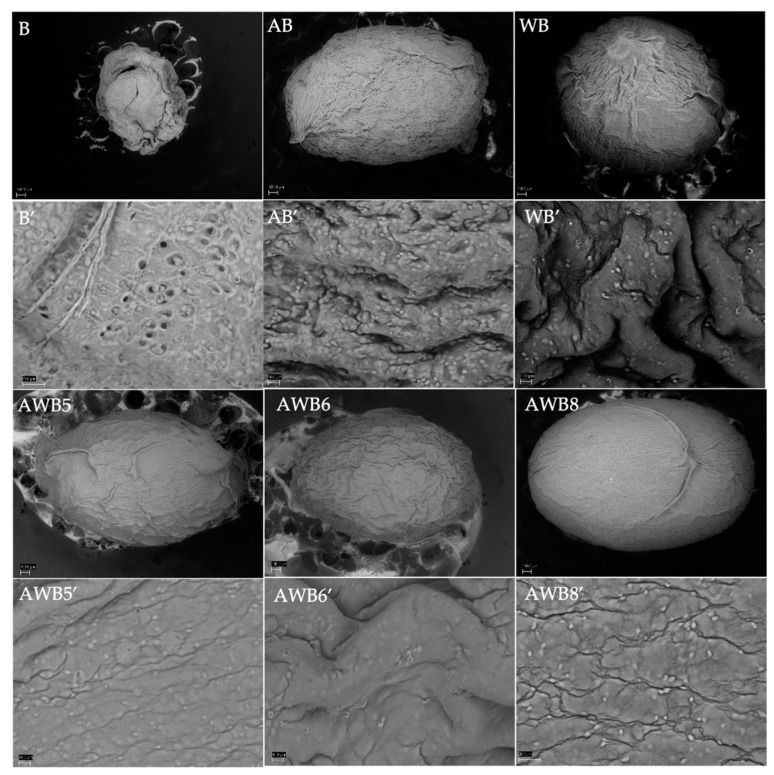
Micrographs obtained by scanning electron microscopy (40x/scale: 100 microns, 500X/scale: 10 microns) of beads with *S. boulardii*. *Nomenclature with apostrophe (’) indicates zoom to 500X.

**Figure 4 bioengineering-09-00460-f004:**
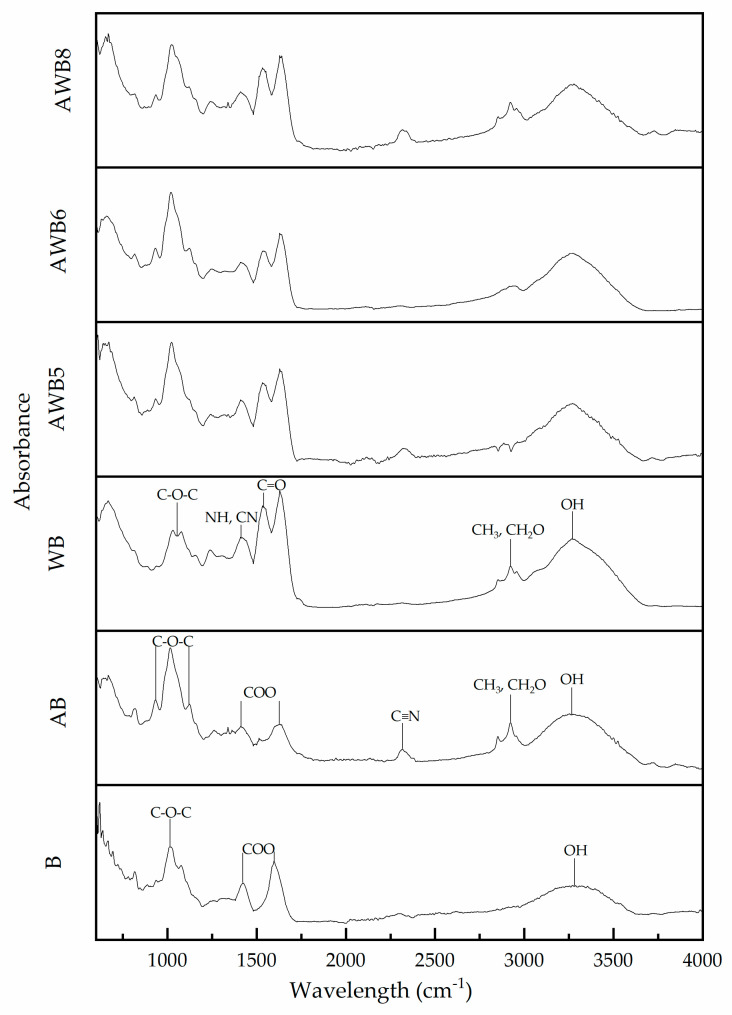
FT-IR absorbance spectra for each of the different beads with *S. boulardii*.

**Figure 5 bioengineering-09-00460-f005:**
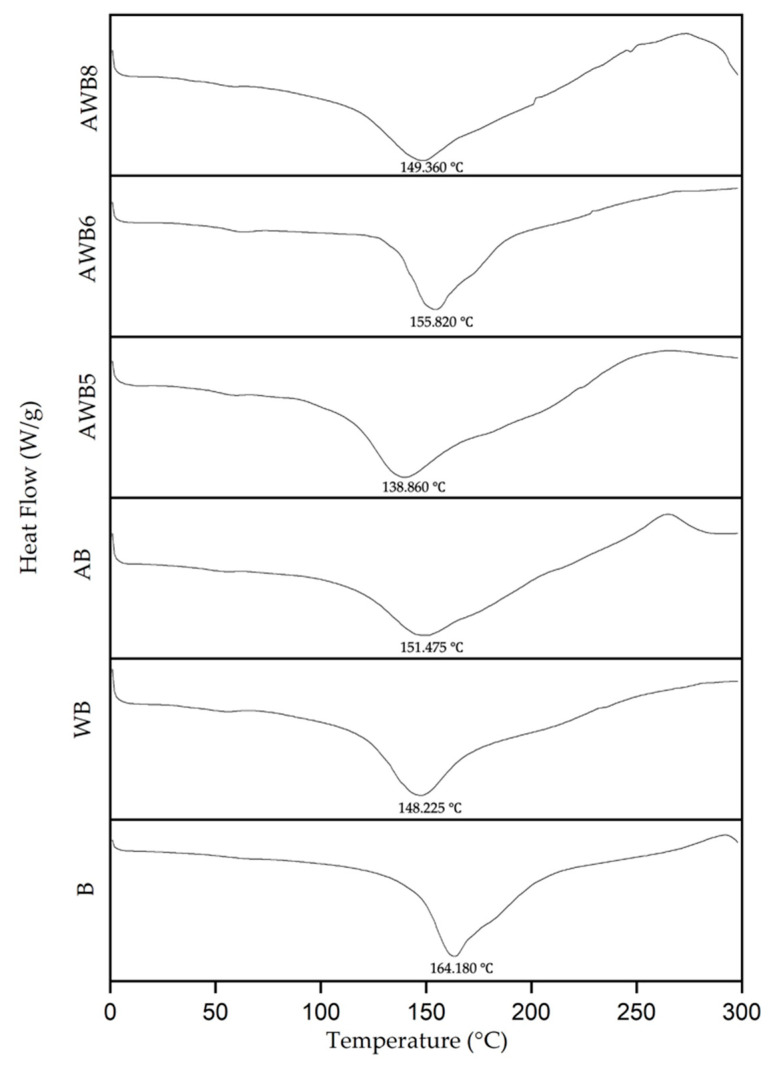
Thermogram of *S. boulardii* encapsulations using agavins and whey protein as wall materials at different concentrations.

**Figure 6 bioengineering-09-00460-f006:**
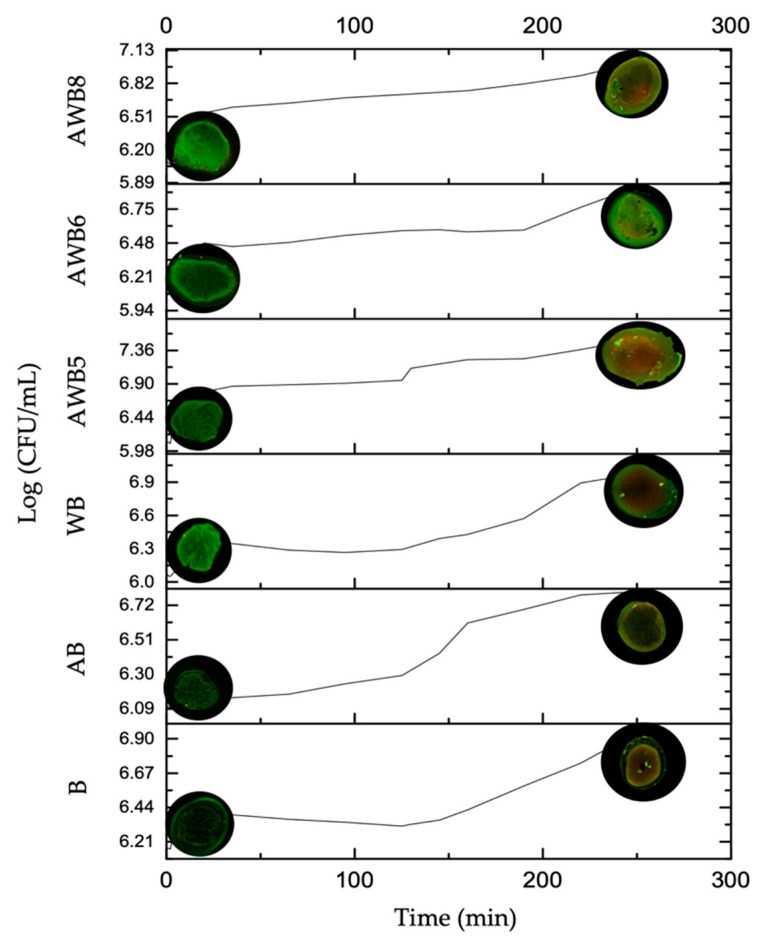
Cell release of *S. boulardii* beads (Log CFU/mL) under in vitro gastrointestinal digestion conditions.

**Figure 7 bioengineering-09-00460-f007:**
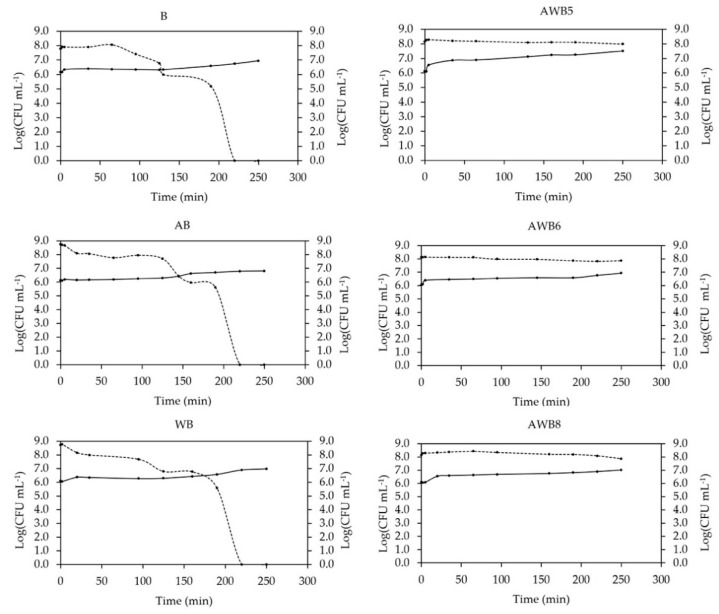
Concentration of *S. boulardii* detected in the medium vs. amount of live *S. boulardii* inside the beads. (line): release of *S. boulardii*, (dotted line): plate count of *S. boulardii*.

**Table 1 bioengineering-09-00460-t001:** Nomenclature and composition of the beads.

Nomenclature	A (%)	WP (%)
B (Control alginate)	0	0
AB (Control Agavins)	5	0
WB (Control WP)	0	5
AWB1	2.5	2.5
AWB2	3.75	2.5
AWB3	5	2.5
AWB4	2.5	3.75
AWB5	3.75	3.75
AWB6	5	3.75
AWB7	2.5	5
AWB8	3.75	5
AWB9	5	5

A: agavins, WP: whey protein.

**Table 2 bioengineering-09-00460-t002:** Release mechanisms and Damköhler number of *S. boulardii* in different beads.

	Korsmeyer–Peppas			Higuchi		Peppas–Sahlin			Release Mechanism of *S. boulardii*	Damköhler Number
	n	k	R^2^	k	R^2^	k_1_	k_2_	R^2^		
B	0.6389	0.0135	0.5025	1.0277	0.4063	−12.429	3.1317	0.8977	Diffusion and relaxation (mainly by relaxation)	0.786
AB	0.1285	0.1650	0.4472	1.4407	0.6039	−0.6832	0.2798	0.8538		0.837
WB	0.3504	0.3354	0.6190	0.8256	0.6926	−10.914	2.8105	0.9848		0.848
AWB5	0.5358	0.0254	0.6852	0.8111	0.7371	−4.5464	1.3314	0.8877		1.063
AWB6	0.4060	0.0757	0.9098	1.3004	0.9144	1.0697	−0.2190	0.9694	Diffusion and relaxation (mainly by diffusion)	1.121
AWB8	0.1987	0.1644	0.9294	1.7422	0.9014	0.3699	−0.0448	0.9315		1.135

## Data Availability

Not applicable.

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
