# Peer review of "Kinetics and Mechanisms of Saccharomyces boulardii Release from Optimized Whey Protein–Agavin–Alginate Beads under Simulated Gastrointestinal Conditions"

_bioengineering, 2022, doi:10.3390/bioengineering9090460_

Round 1
Reviewer 1 Report
The manuscript presented well-designed study to encapsulate probiotics. It has good reader interests. I have some comments to improve the methodology and result presentations as follows:
|
· |
Line/Section |
Comment |
|
· |
Introduction |
More literature is needed about the wall materials used in this study. Additional literature is also needed about the previous work related to the encapsulation of the organism used in this study and what is the gap that this study is trying to fill. |
|
· |
Materials |
Detail out the alginate used in this study: molecular weight and M/G ratio, the source and purity… |
|
· |
(2.6) |
Explain in each section how you [prepared the samples for analysis: wet samples, dried one (lyophilized or how…). |
|
· |
186 |
Add to Carl Zeiss EVO microscope description: scanning electron microscope |
|
· |
213-214 |
There are some replicate time points like 5, 35… |
|
· |
3.3.1 Morphostructural optimization of the beads |
Explain the physical meaning of each parameter like area, perimeter, solidity … |
|
· |
3.3.2 and 3.3.3 |
Regarding the image analysis parameters: add more details about what each parameter represents and what is its importance to the formulation and the yeast. The results were not presented in a table or figure, add the results to the manuscript or the supplement and cite them in the manuscript. The method was also very briefly described, add more details concerning the image taken for analysis, how many beads were analyzed etc. |
|
· |
382 |
Explain what is the fusion temperature observed in the DSC graph. |
|
· |
Figure 6 |
The figure shows the values increases in the y-axis but the viability was mentioned to decrease. Explain. |
|
· |
Release and viability in simulated buffers |
In (2.8.1), explain how you plated the yeasts from the beads. In the results, provide a graph (as supplement will be good) for the cumulative release (as % of the initial amount. It is more representative than the absolute CFU. |
|
· |
465 |
The DSC was not shown to have melting point. |
Author Response
Dear Reviewer:
"Please see the attachment."
PhD. Brenda Camacho Díaz

Reviewer 2 Report
Line 97: Author only says “Wall materials such as whey protein (WP) and reserve polysaccharides such as agavins (A) have been shown to be favorable materials for the encapsulation of probiotic microorganisms and bioactive compounds” in the INTRODUCTION, and has little representation about agavins. The characteristics of whey protein are not be mentioned.
So, the detail advantages of whey protein and agavins should be added in the manuscript.
Line 99: this paper mainly sets out the encapsulation of Saccharomyces boulardii using protein-polysaccharide (as the wall material) and alginate (gelling agent). For agavin, author says agavins have the prebiotic effect and could be use as wall material, but most of polysaccharides could be viewed as wall material and prebiotics. Could the author find relative literature about the prebiotic effect of polysaccharide (agavins) for Saccharomyces boulardii, this will become even more convincing, meanwhile, this could be a creative point. In this manuscript, the innovation was unclear.
Line 111: Author does not state the research progress of encapsulation for probiotic via (whey) protein-polysaccharide wall material, the relative information should be added in the manuscript.
Line 139: how to regulated pH, using HCl or NaOH? The detail should be added in 2.3
Line 215: The absorbance values at the wavelength of 640 nm only indicate the number of Saccharomyces boulardii, however, the survival of Saccharomyces boulardii is not sure. In 2.8.1, viable cell counts were performed based on the plate count method. Why not uniform the method? If most of Saccharomyces boulardii are dead, total research will lose meaning. Author must explain the phenomenon, and add relative experiments for ensuring the survival of cell!
Author Response

(The authors gave the same response as above.)

Reviewer 3 Report
The manuscript describes the effect of the shell material on the survival of Saccharomyces boulardii in an encapsulated form. Every year we have more and more evidence that probiotics improve health in many aspects and are helpful in the prevention of civilization diseases. For this reason, methods of increasing probiotic yeast viability while reaching the colon should be developed.. The manuscript should be published with a few minor corrections.
Page 3, line 99: The chemical characteristics of the agavin should be added, how is the polymer structured, what is the average molecular weight.
Page 12, line 374-379: "The conformational chemical changes observed from the FT-IR spectra (Figure 4) allowed us to identify a (weak) conformational change in the structure of the AB and WB, due to the interaction between alginate-agavins and alginate-whey protein.”
Can you develop the description of this result, the presence of what bonds and in which wavelength range is evidence of interaction?
This result was commented on in the discussion: "Through on DSC and FT-IR analysis, it was found that the mixture of agavins and whey protein maintained and/or compensated the weak interactions that allowed to increase the melting temperature and the energy needed to generate a change in the properties of the compounds that formed the bead.” For this reason, the basis for inference about interactions needs to be clarified.
Figure S2: In the title of the Figure, please add which section of the digestion the S. boulardii microcapsules shown in the micrographs come from.
Author Response

(The authors gave the same response as above.)

Round 2
Reviewer 2 Report
the survival of Saccharomyces boulardii should be studied in your further research